

# Differentiable MadNIS-Lite

Theo Heimel[1], Olivier Mattelaer[2], Tilman Plehn[1,3] and Ramon Winterhalder[2]

**1** Institut für Theoretische Physik, Universität Heidelberg, Germany
**2** CP3, Université catholique de Louvain, Louvain-la-Neuve, Belgium
**3** Interdisciplinary Center for Scientific Computing (IWR), Universität Heidelberg, Germany

## Abstract

Differentiable programming opens exciting new avenues in particle physics, also affecting future event generators. These new techniques boost the performance of current and planned MadGraph implementations. Combining phase-space mappings with a set of very small learnable flow elements, MADNIS-Lite, can improve the sampling efficiency while being physically interpretable. This defines a third sampling strategy, complementing VEGAS and the full MADNIS.

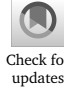
# 1 Introduction

The incredibly successful precision-LHC program is based on comparing vast numbers of scattering events with first-principle predictions provided by multi-purpose event generators, like PYTHIA8 [1], MG5AMC [2], and SHERPA [3]. For a given Lagrangian, they use perturbative quantum field theory to provide the backbone of a complex simulation and inference chain. In view of the upcoming LHC runs, we will have to rely on both modern machine learning (ML) [4, 5] and improved hardware accelerated codes [6–10] to significantly improve the speed and the precision of these simulations. The MADNIS [11, 12] framework is a promising candidate for such ML-based advancements. Following the modular structure of event generators, modern neural networks will transform phase-space sampling [11–22], scattering amplitude evaluations [23–29], event generation [30–36], parton shower generation [37–44], as well as the critical and currently far too slow detector simulations [45–72].

The workhorses for this transformation are generative networks, trained on and amplifying simulated training data [73, 74]. Given the LHC requirements, they have to be controlled and precise in encoding kinematic patterns over an, essentially, interpretable phase space [35, 75–78]. In addition to event generation, these generative networks can be used for event subtraction [79], event unweighting [80, 81], or super-resolution enhancement [82, 83]. Their conditional versions enable new analysis methods, like probabilistic unfolding [84–93], inference [94–97], or anomaly detection [98–103].

An even more advanced strategy to improve LHC simulations is differentiable programming. Here, the entire simulation chain is envisioned to benefit from the availability of derivatives with respect to, for instance, model and tuning parameters in modern computer languages. A proof of principle has been delivered for differentiable matrix elements [104], but the same methods are used for differentiable detector design [105], derivatives of branching processes [106], shower-simulations with path-wise derivatives [107], all the way to a differentiable parton-shower event generator for $e^+e^-$ collisions [108]. The crucial question is where differentiable programming can be used to improve existing ML-enhanced classic event generators.

We will test how a differentiable version of the MADNIS event sampler [11, 12] compares to established improvements which will be part of an upcoming MADGRAPH release for the HL-LHC. We expect our findings to similarly apply to neural importance sampling (NIS) developments in SHERPA [16, 17]. In Sec. 2 we briefly review the MADNIS reference structures, into which we first implement a differentiable integrand, including phase space, matrix element, and parton densities, in Sec. 3. In a second step, we construct a differentiable phase space generator to improve the phase space mapping through a set of small learnable elements, MADNIS-Lite, in Sec. 4. Here we find that differentiable programming can be useful for ML-event generators, but the performance gain has to be benchmarked and evaluated carefully. For MADNIS-Lite, a differentiable phase space generator appears very promising.

# 2 MADNIS basics

We briefly review multi-channel Monte Carlo and the MADNIS basics [11, 12], necessary to understand the subsequent sections. We consider the integral of a function $f \sim |\mathcal{M}|^2$ over phase space

$$I[f] = \int \mathrm{d}x \, f(x), \quad x \in \mathbb{R}^D. \tag{1}$$

It can be decomposed by introducing local channel weights $\alpha_i(x)$ [109, 110]

$$f(x) = \sum_{i=1}^{n_c} \alpha_i(x) f(x), \quad \text{with} \quad \sum_{i=1}^{n_c} \alpha_i(x) = 1, \quad \text{and} \quad \alpha_i(x) \geq 0, \tag{2}$$

using the MG5AMC decomposition. Similar decompositions [111, 112] are used in SHERPA [3] and WHIZARD [113]. The phase-space integral now reads

$$I[f] = \sum_{i=1}^{n_c} \int dx\, \alpha_i(x) f(x). \tag{3}$$

Next, we introduce a set of channel-dependent phase-space mappings

$$x \in \mathbb{R}^D \quad \underset{\leftarrow \overline{G}_i(z)}{\overset{G_i(x) \rightarrow}{\longleftrightarrow}} \quad z \in [0, 1]^D, \tag{4}$$

which parametrize properly normalized densities

$$g_i(x) = \left| \frac{\partial G_i(x)}{\partial x} \right|, \quad \text{with} \quad \int dx\, g_i(x) = 1. \tag{5}$$

The phase-space integral now covers the $D$-dimensional unit hypercube and is sampled as

$$\begin{aligned} I[f] &= \sum_{i=1}^{n_c} \int \frac{dz}{g_i(x)} \alpha_i(x) f(x) \bigg|_{x=\overline{G}_i(z)} \\ &= \sum_{i=1}^{n_c} \int dx\, g_i(x) \frac{\alpha_i(x) f(x)}{g_i(x)} \equiv \sum_{i=1}^{n_c} \left\langle \frac{\alpha_i(x) f(x)}{g_i(x)} \right\rangle_{x \sim g_i(x)}. \end{aligned} \tag{6}$$

This is the basis of multi-channel importance sampling [112]. Starting from this equation, MADNIS encodes the multi-channel weight $\alpha_i(x)$ and the channel mappings $G_i(x)$ in neural networks.

**Neural channel weights**

First, MADNIS employs a channel-weight network to encode the local multi-channel weights

$$\alpha_i(x) \equiv \alpha_{i\xi}(x), \tag{7}$$

where $\xi$ denotes the network parameters. As the channel weights vary strongly over phase space, it helps the performance to learn them as a correction to a physically motivated prior assumption like [109, 110]

$$\begin{aligned} \alpha_i^{\text{MG,1}}(x) &= \frac{|\mathcal{M}_i(x)|^2}{\sum_j |\mathcal{M}_j(x)|^2}, \quad \text{or} \\ \alpha_i^{\text{MG,2}}(x) &= \frac{P_i(x)}{\sum_j P_j(x)}, \quad \text{with} \quad P_i(x) = \prod_{k \in \text{prop}} \frac{1}{|p_k(x)^2 - m_k^2 - i m_k \Gamma_k|^2}. \end{aligned} \tag{8}$$

Relative to either of them, we then only learn a correction factor [12].

**Neural importance sampling**

Second, MADNIS combines analytic channel mappings with a normalizing flow [114, 115],

$$x \in \mathbb{R}^D \xleftrightarrow{\text{analytic}} y \in [0,1]^D \xleftrightarrow{\text{flow}} z \in [0,1]^D, \tag{9}$$

to complement VEGAS [116–118]. The flow allows MADNIS to improve the physics-inspired phase-space mappings by training a network to map

$$z = G_{i\theta}(x), \quad \text{or} \quad x = \overline{G}_{i\theta}(z), \tag{10}$$

where $\theta$ denotes another set of network parameters. As for the channel weights, we use physics knowledge to simplify the ML task and a VEGAS pre-training [12]. This makes use of the key strength of VEGAS, which is extremely efficient for factorizing integrands and converges much faster than neural importance sampling.

**Multi-channel variance loss**

With the channel weights and importance sampling encoded in neural networks,

$$\alpha_i(x) \equiv \alpha_{i\xi}(x), \quad \text{and} \quad g_i(x) \equiv g_{i\theta}(x), \tag{11}$$

the integral in Eq.(6) becomes

$$I[f] = \sum_{i=1}^{n_c} \left\langle \frac{\alpha_{i\xi}(x)f(x)}{g_{i\theta}(x)} \right\rangle_{x \sim g_{i\theta}(x)}. \tag{12}$$

Crucially, we then minimize the variance as the loss [11, 12]

$$
\begin{aligned}
\mathcal{L}_{\text{variance}} &= \sum_{i=1}^{n_c} \frac{N}{N_i} \sigma_i^2 \\
&= \sum_{i=1}^{n_c} \frac{N}{N_i} \left( \left\langle \frac{\alpha_{i\xi}(x)^2 f(x)^2}{g_{i\theta}(x) q_i(x)} \right\rangle_{x \sim q_i(x)} - \left\langle \frac{\alpha_{i\xi}(x)f(x)}{q_i(x)} \right\rangle^2_{x \sim q_i(x)} \right).
\end{aligned}
\tag{13}
$$

In practice, $q_i(x) \simeq g_{i\theta}(x)$ allows us to compute the loss as precisely as possible and stabilizes the combined online [119] and buffered training [11]. Inspired by stratified sampling, we encode the optimal choice for $N_i$ [112, 120] in the MADNIS loss [12]

$$
\begin{aligned}
\mathcal{L}_{\text{MADNIS}} &= \sum_{i=1}^{n_c} \left( \sum_{j=1}^{n_c} \sigma_j \right) \sigma_i = \left[ \sum_{i=1}^{n_c} \sigma_i \right]^2 \\
&= \left[ \sum_{i=1}^{n_c} \left( \left\langle \frac{\alpha_{i\xi}(x)^2 f(x)^2}{g_{i\theta}(x) q_i(x)} \right\rangle_{x \sim q_i(x)} - \left\langle \frac{\alpha_{i\xi}(x)f(x)}{q_i(x)} \right\rangle^2_{x \sim q_i(x)} \right)^{1/2} \right]^2.
\end{aligned}
\tag{14}
$$

## 3 Differentiable integrand

To investigate how the choice of the loss function and the direction of the training affects the performance of MADNIS, we consider a realistic LHC process, namely triple-W production,

$$u\bar{d} \to W^+ W^+ W^-. \tag{15}$$

At leading order, this process comes with 17 Feynman diagrams and 16 integration channels in MG5AMC. It has been shown to benefit significantly from ML-based importance sampling methods [12] and that a single fine-tuned integration channel is sufficient to achieve good performance. We have implemented a simple differentiable event generator for this process in MADNIS, including

- a differentiable squared matrix element using helicity amplitudes natively written in PY-TORCH that are generated from our custom MG5AMC plugin similar to MADFLOW [121] and MADJAX [104],

- a differentiable and invertible phase-space generator based on RAMBOONDIET [76, 122, 123] natively written in PYTORCH. A similar implementation has been used in MAD-JAX [104],

- and differentiable parton densities using the standard LHAPDF interpolation [124] natively written in PYTORCH which is similar to PDFFLOW [125] relying on TENSORFLOW.

### 3.1 Forward and inverse training

As a starting point we consider a generic $F$-divergence [126] between two normalized probability distributions $p_1(z)$ and $p_2(z)$,

$$D_F^z[p_1, p_2] = \int \mathrm{d}z \, p_2(z) \, F\left(\frac{p_1(z)}{p_2(z)}\right). \tag{16}$$

For a MADNIS-like training, we have a target function $f(x)$ and a normalizing flow that parametrizes a trainable invertible mapping with network parameters $\theta$

$$x \quad \underset{\leftarrow \overline{G}_\theta(z)}{\overset{G_\theta(x) \rightarrow}{\longleftrightarrow}} \quad z, \tag{17}$$

and induces the density distribution

$$g_\theta(x) = p_0(G_\theta(x)) \left| \frac{\partial G_\theta(x)}{\partial x} \right|, \tag{18}$$

with latent space distribution $p_0(z)$. For convenience, we can also define

$$\overline{g}_\theta(z) = p_0(z) \left| \frac{\partial \overline{G}_\theta(z)}{\partial z} \right|^{-1}, \quad \text{such that} \quad \overline{g}_\theta(G_\theta(x)) = g_\theta(x). \tag{19}$$

Note that $g_\theta(x)$ is a normalized probability distribution in $x$-space (data space), but this is not the case for $\overline{g}_\theta(z)$ in $z$-space (latent space). There are two ways to define a loss function to train the flow. First, we define the loss function in data space,

$$\mathcal{L}_F^{\mathrm{fw}} = D_F^x[f, g_\theta] = \int \mathrm{d}x \, g_\theta(x) \, F\left(\frac{f(x)}{g_\theta(x)}\right) = \left\langle \frac{g_\theta(x)}{q(x)} F\left(\frac{f(x)}{g_\theta(x)}\right) \right\rangle_{x \sim q(x)}. \tag{20}$$

In the last step, we introduce an importance sampling distribution $q(x)$ to evaluate the integral numerically. This can be the same as $g_\theta(x)$ for online training, or different for buffered training [12]. Optimizing this loss function requires evaluating the flow in the forward direction according to Eq.(18), so we refer to this training mode as forward training.

Alternatively, we can train in latent space using the remapped target distribution

$$\hat{f}(z) = f\left(\overline{G}_\theta(z)\right) \left| \frac{\partial \overline{G}_\theta(z)}{\partial z} \right|, \tag{21}$$

which is a normalized probability in latent space according to the change of variables formula. During training we minimize the divergence between $\hat{f}(z)$ and the latent space distribution

$$
\begin{aligned}
\mathcal{L}_F^{\text{inv}} = D_F^z[\hat{f}, p_0] &= \int \mathrm{d}z\, p_0(z) F\left(\frac{\hat{f}(z)}{p_0(z)}\right) \\
&= \int \mathrm{d}z\, p_0(z) F\left(\frac{f(\overline{G}_\theta(z))}{p_0(z)}\left|\frac{\partial \overline{G}_\theta(z)}{\partial z}\right|\right) \\
&= \int \mathrm{d}z\, p_0(z) F\left(\frac{f(\overline{G}_\theta(z))}{\overline{g}_\theta(z)}\right) = \left\langle F\left(\frac{f(\overline{G}_\theta(z))}{\overline{g}_\theta(z)}\right)\right\rangle_{z\sim p_0(z)}.
\end{aligned}
\tag{22}
$$

We refer to this as inverse training because we evaluate the flow in the inverse direction. Note that this inverse training requires a differentiable integrand which is not needed in the forward training. It turns out that the forward and inverse training yield the same result for the loss,

$$
\begin{aligned}
\mathcal{L}_F^{\text{inv}} &= \int \mathrm{d}x\, p_0(G_\theta(x))\left|\frac{\partial G_\theta(x)}{\partial x}\right| F\left(\frac{f(\overline{G}_\theta(G_\theta(x)))}{\overline{g}_\theta(G_\theta(x))}\right) \\
&= \int \mathrm{d}x\, g_\theta(x) F\left(\frac{f(x)}{g_\theta(x)}\right) = \mathcal{L}_F^{\text{fw}}.
\end{aligned}
\tag{23}
$$

**Loss gradients**

Because the two losses are identical, the gradients of the forward and inverse loss functions have the same expectation value,

$$
\nabla_\theta \mathcal{L}_F^{\text{inv}} = \left\langle \nabla_\theta F\left(\frac{f(\overline{G}_\theta(z))}{\overline{g}_\theta(z)}\right)\right\rangle_{z\sim p_0(z)} = \left\langle \nabla_\theta \frac{g_\theta(x)}{q(x)} F\left(\frac{f(x)}{g_\theta(x)}\right)\right\rangle_{x\sim q(x)} = \nabla_\theta \mathcal{L}_F^{\text{fw}}.
\tag{24}
$$

From these expressions we can see that the two methods are not equivalent for the variances of the gradients. For the inverse training, it is

$$
\begin{aligned}
\mathrm{Var}_{z\sim p_0(z)}\left(\nabla_\theta F\left(\frac{f(\overline{G}_\theta(z))}{\overline{g}_\theta(z)}\right)\right) &= \left\langle \left(\nabla_\theta F\left(\frac{f(\overline{G}_\theta(z))}{\overline{g}_\theta(z)}\right) - \nabla_\theta \mathcal{L}_F^{\text{inv}}\right)^2\right\rangle_{z\sim p_0(z)} \\
&= \int \mathrm{d}z\, p_0(z)\left(\nabla_\theta F\left(\frac{f(\overline{G}_\theta(z))}{\overline{g}_\theta(z)}\right) - \nabla_\theta \mathcal{L}_F^{\text{inv}}\right)^2,
\end{aligned}
\tag{25}
$$

while for the forward direction, we find

$$
\begin{aligned}
\mathrm{Var}_{x\sim q(x)}\left(\nabla_\theta \frac{g_\theta(x)}{q(x)} F\left(\frac{f(x)}{g_\theta(x)}\right)\right) &= \left\langle \left(\nabla_\theta \frac{g_\theta(x)}{q(x)} F\left(\frac{f(x)}{g_\theta(x)}\right) - \nabla_\theta \mathcal{L}_F^{\text{fw}}\right)^2\right\rangle_{x\sim q(x)} \\
&= \int \mathrm{d}x\, q(x)\left(\nabla_\theta \frac{g_\theta(x)}{q(x)} F\left(\frac{f(x)}{g_\theta(x)}\right) - \nabla_\theta \mathcal{L}_F^{\text{fw}}\right)^2.
\end{aligned}
\tag{26}
$$

In general, the two will not be equal. Depending on the problem and choice of $F$-divergence, one of the two training modes can have noisier gradients, leading to a slower convergence of the training on finite batch sizes. In Appendix B, we study this behavior by analytically solving a simple 1D-toy example.

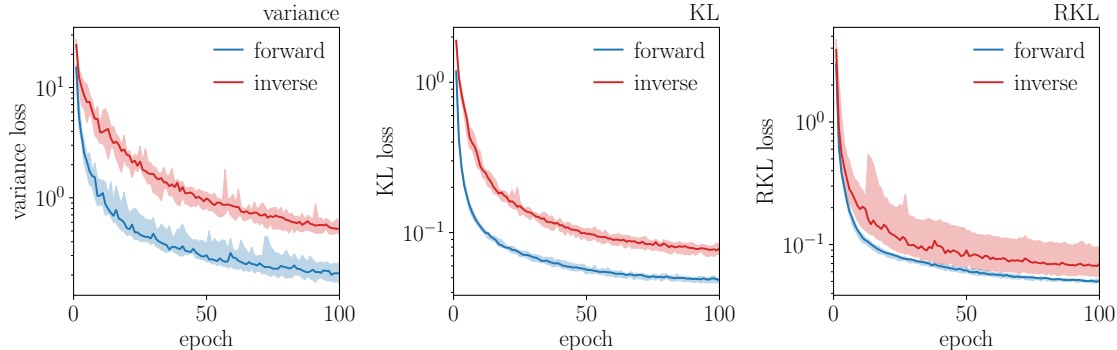

Figure 1: Loss values as a function of the training epochs for different losses: variance, KL-divergence, and RKL-divergence from left to right.

## 3.2 Loss landscape

Finally, we look at established examples of divergences used to construct loss functions. They use a normalized target distribution $f(x)$, which can be ensured during training by using batch-wise normalization of the integrand values:

- variance, $F(t) = (t-1)^2$

$$\mathcal{L}_{\text{var}}^{\text{fw}} = \left\langle \frac{g_\theta(x)}{q(x)} \left( \frac{f(x)}{g_\theta(x)} - 1 \right)^2 \right\rangle_{x \sim q(x)},$$

$$\mathcal{L}_{\text{var}}^{\text{inv}} = \left\langle \left( \frac{f(\overline{G}_\theta(z))}{\overline{g}_\theta(z)} - 1 \right)^2 \right\rangle_{z \sim p_0(z)}. \tag{27}$$

- KL-divergence, $F(t) = t \log t$

$$\mathcal{L}_{\text{KL}}^{\text{fw}} = \left\langle \frac{f(x)}{q(x)} \log \frac{f(x)}{g_\theta(x)} \right\rangle_{x \sim q(x)},$$

$$\mathcal{L}_{\text{KL}}^{\text{inv}} = \left\langle \frac{f(\overline{G}_\theta(z))}{\overline{g}_\theta(z)} \log \frac{f(\overline{G}_\theta(z))}{\overline{g}_\theta(z)} \right\rangle_{z \sim p_0(z)}. \tag{28}$$

- reverse KL-divergence, $F(t) = -\log t$

$$\mathcal{L}_{\text{RKL}}^{\text{fw}} = \left\langle \frac{g_\theta(x)}{q(x)} \log \frac{g_\theta(x)}{f(x)} \right\rangle_{x \sim q(x)},$$

$$\mathcal{L}_{\text{RKL}}^{\text{inv}} = \left\langle \log \frac{\overline{g}_\theta(z)}{f(\overline{G}_\theta(z))} \right\rangle_{z \sim p_0(z)}. \tag{29}$$

We can test these six scenarios of forward and inverse training using three different divergences on triple-W production. To this end, we run a simple single-channel MADNIS training without buffered training, with the hyperparameters given in Tab. 2. We estimate the stability of the training and results by repeating each training ten times. In Fig. 1, we first show the training behavior for the different scenarios. We immediately see that the KL-divergence leads to the most stable training. In contrast, the RKL-divergence combined with inverse training is the most noisy. In terms of performance measured by the final loss value, forward training consistently beats inverse training.

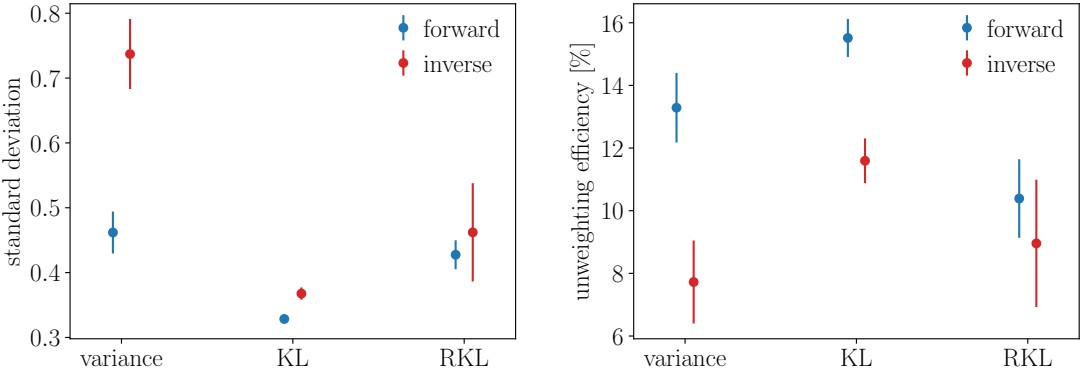

Figure 2: Relative standard deviations (left) and unweighting efficiencies (right) for different loss functions.

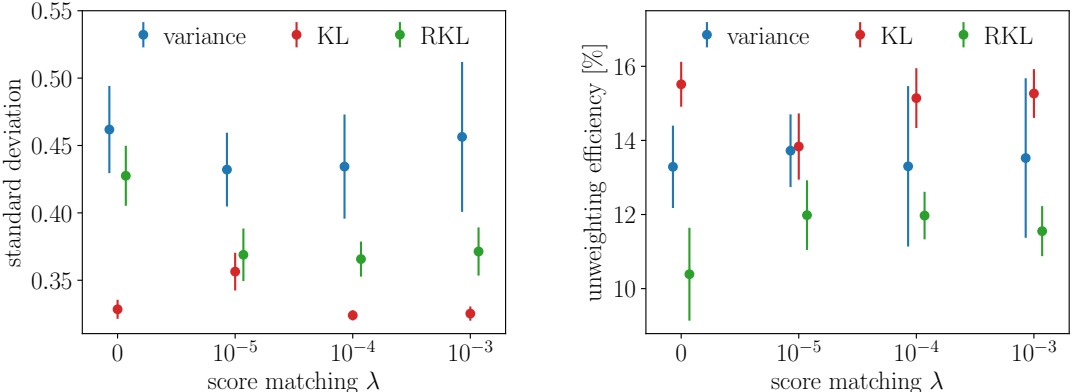

Figure 3: Relative standard deviations (left) and unweighting efficiencies (right) for different derivative matching coefficients.

In Fig. 2, we show the unweighting efficiencies and the standard deviations of the integral as more quantitative quality measures. For the standard deviation, we see that variance and RKL losses used for forward training lead to similar results, but the KL-divergence outperforms them. Also, in terms of unweighting efficiency, forward training with a KL-loss leads to the best results. The fact that RKL gives the worst unweighting efficiency is related to overweights, which the RKL does not penalize. This comparison has to be taken with a grain of salt, because the performance of forward training based on the variance and the KL-divergence are close in performance. An additional aspect we have to factor in is that a multi-channel loss can only be constructed using the variance, whereas the KL-divergence might be most suitable for single-channel integrals.

**Additional derivatives**

When using differentiable integrands, we can also evaluate an additional derivative matching term (also called score or force matching [104]) for each forward loss introduced above,

$$\mathcal{L}^{\text{fw}} \rightarrow \mathcal{L}^{\text{fw}} + \lambda \left\langle |\partial_x \log f(x) - \partial_x \log g_\theta(x)|^2 \right\rangle_{x \sim q(x)} . \tag{30}$$

The relative strength of the derivative term, $\lambda$, is a hyperparameter. In Fig. 3, we show the same triple-W results as in Fig. 2, but including derivative matching with different strengths $\lambda$. For the variance and RKL losses, we see slight improvements in the results from the derivative matching. However, it turns out that it comes with less stable training. Altogether, the

improvements from adding the derivative are relatively small, leading to the non-trivial question if their performance gain justifies the additional computational cost from evaluating the gradients. Moreover, generalizing this additional term to buffered training for multi-channel integration would come with a very large memory footprint.

# 4 Differentiable phase space — MADNIS-Lite

To define differentiable and trainable phase-space mappings, we re-introduce the relevant building blocks using consistent notation. For a hadron-collider process

$$p_1 + p_2 \rightarrow k_1 + \ldots + k_n\,, \tag{31}$$

the differential cross section is given by

$$d\sigma = \sum_{a,b} dx_1\, dx_2\, d\Phi^{2\rightarrow n}\, f_a(x_1) f_b(x_2) \frac{(2\pi)^{4-3n}}{2x_1 x_2 s}\, |\mathcal{M}_{ab}(p_1, p_2 | k_1, \ldots, k_n)|^2\,, \tag{32}$$

with a sum over initial-state partons and the phase space density

$$d\Phi^{2\rightarrow n} = \left[ \prod_{i=1}^{n} d^4 k_i\, \delta(k_i)\, \theta(k_i^0) \right] \delta^{(4)}\left( p_1 + p_2 - \sum_i k_i \right)\,. \tag{33}$$

The total cross section is

$$\sigma = \int dx_1\, dx_2\, d\Phi^{2\rightarrow n}(x)\, f(x) \equiv \int dx\, f(x)\,, \tag{34}$$

$$\text{with} \quad f(x) = \frac{(2\pi)^{4-3n}}{2x_1 x_2 s} \sum_{a,b} f_a(x_1) f_b(x_2)\, |\mathcal{M}_{ab}(x)|^2\,, \tag{35}$$

in terms of the phase space vector $x = (x_1, x_2, k_i)$.

## 4.1 Parameterized mappings

For a single channel, we choose a suitable mapping of $x$ to a unit-hypercube $z$,

$$\sigma = \int dx\, f(x) = \int dz\, \left. \frac{f(x)}{g(x)} \right|_{x=\overline{G}(z)}\,. \tag{36}$$

To minimize the integration error, we choose the mapping $g$ to follow the peaking propagator structure of $f(x)$, as encoded in the Feynman diagrams. To this end, we decompose the phase-space integral into integrals over time-like invariants $s_i$, $2 \rightarrow 2$ scattering processes with $t$-channel propagators, and $1 \rightarrow 2$ particle decays,

$$\int d\Phi^{2\rightarrow n}(x) = \prod_{i=1}^{n-2} \int_{s_{i,\min}}^{s_{i,\max}} ds_i \prod_{j=1}^{\kappa} \int d\Phi_j^{2\rightarrow 2}(x) \prod_{k=1}^{n-\kappa-1} \int d\Phi_k^{1\rightarrow 2}(x)\,. \tag{37}$$

The number of $2 \rightarrow 2$ processes and $1 \rightarrow 2$ decays depends on the diagram the mapping is based on. If several $t$-channel propagators are present, i.e. for $\kappa \geq 2$, some of the $s_i$ do not correspond to propagators in the diagram and they are sampled uniformly. For the $(3n-4)$-dimensional integral this results in

- $d_s = n - 2$ degrees of freedom from time-like invariants,

- $d_p = 2\kappa$ degrees of freedom from $2 \to 2$ scatterings,

- $d_d = 2(n - \kappa - 1)$ degrees of freedom from decays.

Together with the PDF convolutions, this covers $3n - 2$ integral dimensions. For each of these sub-integrals, it is possible to define appropriate mappings $x \leftrightarrow z$ as it is commonly done in many multi-purpose event generators [1–3]. We implement these physics-inspired mappings in a differentiable and invertible way using PYTORCH, allowing us to perform forward training. In the following, we briefly review our parametrization [127] for each phase-space block needed to understand how they can be upgraded by additional trainable transformations.

**Propagator invariants**

To construct a smooth mapping for a propagator

$$|\mathcal{M}|^2 \propto \frac{1}{(s - M^2)^2 + M^2 \Gamma^2}, \tag{38}$$

we map the invariant $s$ to a random number $z_s = G_{\text{prop}}(s)$ such that

$$\int_{s_{\min}}^{s_{\max}} \mathrm{d}s = \int_0^1 \frac{\mathrm{d}z}{g_{\text{prop}}(s(z_s), s_{\min}, s_{\max})} = \int_0^1 \mathrm{d}z \left| \frac{\partial \overline{G}_{\text{prop}}(z_s, m^2, s_{\min}, s_{\max})}{\partial z} \right|. \tag{39}$$

Depending on the propagator width, we can introduce two different mappings. For a Breit-Wigner propagator we use

$$\overline{G}_{\text{prop}}^{\text{BW}}(z_s, m^2, s_{\min}, s_{\max}) = m\Gamma \tan[y_1 + (y_2 - y_1)z_s] + m^2,$$

$$g_{\text{prop}}^{\text{BW}}(s, m^2, s_{\min}, s_{\max}) = \frac{m\Gamma}{(y_2 - y_1)[(s - m^2)^2 + m^2 \Gamma^2]}, \tag{40}$$

$$y_{1/2} = \arctan\left(\frac{s_{\min/\max} - m^2}{m\Gamma}\right).$$

For $\Gamma = 0$ we instead employ

$$\overline{G}_{\text{prop}}^{\nu}(z_s, m^2, s_{\min}, s_{\max}) = \left[z_s(s_{\max} - m^2)^{1-\nu} + (1 - z_s)(s_{\min} - m^2)^{1-\nu}\right]^{\frac{1}{1-\nu}} + m^2,$$

$$g_{\text{prop}}^{\nu}(s, m^2, s_{\min}, s_{\max}) = \frac{1 - \nu}{[(s_{\max} - m^2)^{1-\nu} - (s_{\min} - m^2)^{1-\nu}](s - m^2)^{\nu}}, \tag{41}$$

as long as $\nu \neq 1$. The parameter $\nu$ can be tuned, and the naive expectation $\nu = 2$ is not necessarily the best choice. We choose $\nu = 1.4$ and then optimize the mapping as explained below.

**$2 \to 2$ scattering processes**

For a $2 \to 2$ scattering with $p_1 + p_2 = k_1 + k_2$, the momenta $p_{1,2}$ and the virtualities $k_i^2$ are fixed or sampled from other phase-space components. As this includes a $t$-channel propagator,

we choose

$$\int d\Phi^{2\to2}(p_1,p_2;k_1^2,k_2^2) = \frac{1}{4\sqrt{\lambda(p^2,p_1^2,p_2^2)}} \int_0^{2\pi} d\phi^* \int_{-t_{\max}}^{-t_{\min}} d|t|$$

$$= \int_0^1 \frac{dz_\phi \, dz_t}{g_{2\to2}(p^2,p_1^2,p_2^2,t,m^2,\nu,t_{\min},t_{\max})},$$

(42)

where $\phi^*$ is the azimuthal angle defined by $p_1$ and $k_1$ in the CM frame of $p = p_1 + p_2$, and $\lambda(x,y,z) = (x-y-z)^2 - 4yz$. The invariant $t = (p_1 - k_1)^2 < 0$ depends only linearly on the azimuthal angle $\cos\theta^*$ as

$$t = k_1^2 + p_1^2 - \frac{(p^2+k_1^2-k_2^2)(p^2+p_1^2-p_2^2) - \sqrt{\lambda(p^2,k_1^2,k_2^2)}\sqrt{\lambda(p^2,p_1^2,p_2^2)}\cos\theta^*}{2p^2}.$$

(43)

The integration boundaries can be calculated from this with $-1 \le \cos\theta^* \le 1$. We sample the polar angle and $t$ according to

$$\phi^* = 2\pi z_\phi, \quad \text{and} \quad |t| = \overline{G}_{\text{prop}}^\nu(z_t,m^2,-t_{\max},-t_{\min}),$$

(44)

with, correspondingly,

$$g_{2\to2}(p^2,p_1^2,p_2^2,t,m^2,\nu,t_{\min},t_{\max}) = \frac{2}{\pi}\sqrt{\lambda(p^2,p_1^2,p_2^2)}\,g_{\text{prop}}(-t,m^2,\nu,-t_{\max},-t_{\min}).$$

(45)

Further, for each $t$-channel block in our diagram, the corresponding $s$-invariant, i.e. $p^2 = s$, also needs to be sampled as time-like invariant in Eq.(37). However, in contrast to invariants reflecting propagators in the diagram, these invariants belong to pseudo particles and can be sampled flat

$$\overline{G}_{\text{pseudo}}(z_s,s_{\min},s_{\max}) = z_s\,(s_{\max}-s_{\min}) + s_{\min},$$

$$g_{\text{pseudo}}(s,s_{\min},s_{\max}) = \frac{1}{s_{\max}-s_{\min}}.$$

(46)

## $1 \to 2$ particle decays

For isotropic decays with $p = k_1 + k_2$, the momentum $p$ and the virtualities $k_i^2$ are again sampled from other phase-space components. We choose the polar angle $\phi^*$ and azimuthal angle $\theta^*$ in the decay rest frame as integration variables and sample $\phi^* = 2\pi z_\phi$ and $\cos\theta^* = 2z_\theta - 1$ uniformly,

$$\int d\Phi^{1\to2}(p;k_1^2,k_2^2) = \frac{\sqrt{\lambda(p^2,k_1^2,k_2^2)}}{8p^2} \int_0^{2\pi} d\phi^* \int_{-1}^1 d\cos\theta^* = \frac{1}{g_{\text{decay}}(p^2,k_1^2,k_2^2)} \int_0^1 dz_1 dz_2,$$

$$\text{with} \quad g_{\text{decay}}(p^2,k_1^2,k_2^2) = \frac{2p^2}{\pi\sqrt{\lambda(p^2,k_1^2,k_2^2)}}.$$

(47)

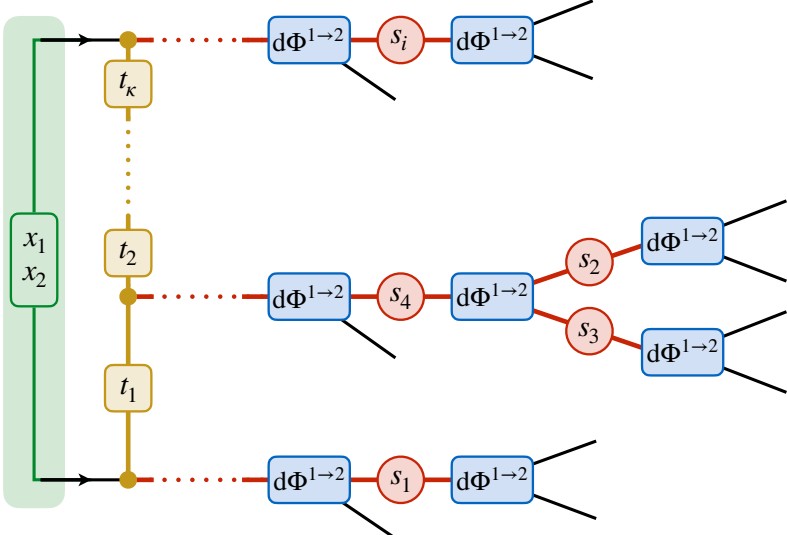

Figure 4: Topological diagram illustrating our separable and differentiable phase-space mappings. Each colored block represents one of the introduced components which can be modified by a trainable bilinear flow.

**PDF convolutions**

For the PDF convolutions, we introduce $\tau = x_1 x_2$, such that the squared partonic CM energy is given by $\hat{s} = \tau s$. This allows us to write

$$\int_0^1 dx_1 dx_2 \, \Theta(\hat{s} - \hat{s}_{\min}) = \int_{\tau_{\min}}^1 d\tau \int_\tau^1 \frac{dx_1}{x_1} = \int_0^1 \frac{dz_\tau dz_{x_1}}{g_{\text{lumi}}(\tau, \tau_{\min})}, \tag{48}$$

$$\text{with} \quad g_{\text{lumi}}(\tau, \tau_{\min}) = \frac{1}{\tau \, \log \tau \, \log \tau_{\min}},$$

where $\hat{s}_{\min}$ follows from final-state masses and cuts and we sample

$$\tau = \tau_{\min}^{1-z_\tau}, \quad \text{and} \quad x_1 = \tau^{z_{x_1}}. \tag{49}$$

The induced density $g_{\text{lumi}}$ exactly cancels the flux factor $\tau^{-1}$ in Eq.(32). If there are no $t$-channels, i.e. $\kappa = 0$, the squared CM energy $\hat{s}$ also belongs to a propagator in the diagram. In this case, it is beneficial to sample $\tau$ such that this propagator structure is mapped out.

Each of the $s$-invariants, $2 \to 2$ scatterings, and decay blocks described above transform one or two random numbers. They can appear multiple times for a given Feynman diagram, as illustrated in Fig. 4. In Appendix C, we illustrate how these components are combined to parametrize a complete channel mapping for $W + 4$ jets production.

## 4.2 Learnable bilinear spline flows

For a typical MADNIS training, the flow sub-networks often encode relatively simple functions. For these cases, we introduce bilinear spline flows to replace the sub-networks with second-order polynomials. A $d_x$-dimensional transformation $x \leftrightarrow z$ with a $d_c$-dimensional condition $c$ can be written as

$$z = G(x; W\hat{c}), \quad \text{with} \quad \hat{c} = \begin{pmatrix} 1 \\ c_i \\ c_i c_j \end{pmatrix}, \quad \text{for} \quad i \le j, \tag{50}$$

Table 1: Trainable components.

| Mapping | Parameters | Conditions |
|---|---|---|
| Time-like invariants, Eqs.(40),(41) (separate for massless and massive propagators) | 190 | partonic CM energy $\sqrt{\hat{s}/s_{\text{lab}}}$ <br> minimal decay CM energy $\sqrt{s_{\min}/s_{\text{lab}}}$ <br> maximal decay CM energy $\sqrt{s_{\max}/s_{\text{lab}}}$ |
| $2 \to 2$ scattering, Eq.(44) | 798 | correlations between $z_t, z_\phi$ <br> partonic CM energy $\sqrt{\hat{s}/s_{\text{lab}}}$ <br> scattering CM energy $\sqrt{p^2/s_{\text{lab}}}$ <br> virtualities $\sqrt{k_{1,2}^2/s_{\text{lab}}}$ |
| Time-like invariants for pseudo-particles, Eq.(46) | 190 | partonic CM energy $\sqrt{\hat{s}/s_{\text{lab}}}$ <br> minimal energy $\sqrt{s_{\min}/s_{\text{lab}}}$ <br> maximal energy $\sqrt{s_{\max}/s_{\text{lab}}}$ |
| $1 \to 2$ decay, Eq.(47) | 380 | correlations between $z_\theta, z_\phi$ <br> partonic CM energy $\sqrt{\hat{s}/s_{\text{lab}}}$ <br> decay CM energy $\sqrt{p^2/s_{\text{lab}}}$ |
| PDF convolutions, Eq.(49) | 114 | correlations between $z_\tau, z_{x_1}$ |

where $G$ is a rational quadratic spline transformation and $W$ is a trainable matrix. The number of trainable parameters for such a transformation with $n_b$ bins is

$$d_W = (3n_b + 1) \times d_x \times \left(1 + d_c + \frac{1}{2} d_c (d_c + 1)\right). \tag{51}$$

This way, we can build small and fast, but sufficiently expressive trainable transformations for a small number of dimensions $d_x$ and $d_c$. Another benefit is the interpretability of bilinear spline flows because $W$ tells us how strongly the spline transformation is correlated with the conditional inputs.

We can combine these trainable mappings with the propagator, decay, scattering, and PDF blocks introduced above and use them to transform their uniform random number input. For mappings with two random numbers, we allow for correlations between the two dimensions. Because all parts of the phase-space mappings are differentiable, the bilinear flow can even be conditional on intermediate physical features that are available only during the evaluation of the phase-space mapping. This enhances the expressivity and interpretability of the learned transformation.

**Implementation**

We implement the trainable bilinear spline flows with 6 spline bins. We list the trainable components of the phase space mappings, the conditional features, and the number of trainable parameters in Tab. 1. These parameters are shared between channels and multiple instances of the same block in one channel. This way, the number of trainable parameters stays the same for different processes and allows the use of mappings trained on one process, like $W + 3$ jets, for reasonably related other processes. Note that for processes like $W + 4$ jets ($t\bar{t} + 3$ jets) with up to 384 (945) integration channels, this parameter sharing reduces the computational cost significantly.

We train MADNIS-Lite using the multi-channel variance loss from Eq.(14), but without trainable channel weights. We use stratified training to focus the available training samples on channels with a large contribution to the total cross section, and buffered training to reduce the number of integrand evaluations. The training hyperparameters are given in Tab. 3.

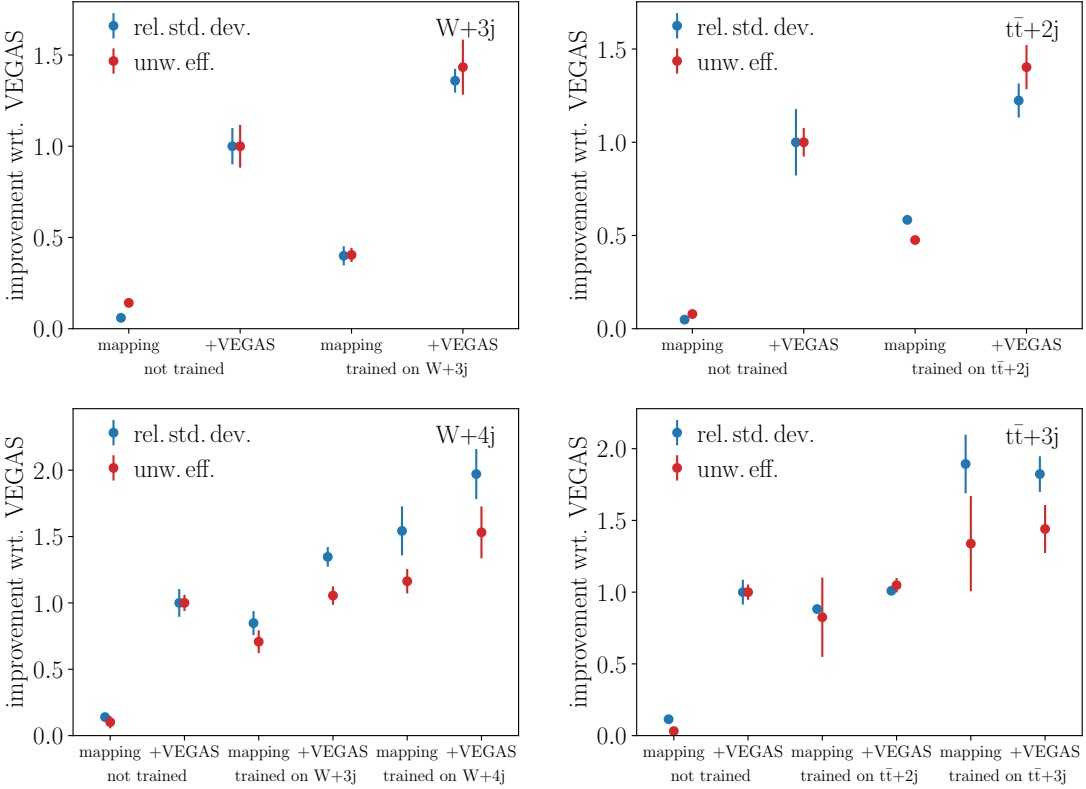

Figure 5: Improvement of the unweighting efficiency and relative standard deviation of different setups with respect to an untrained phase space mapping refined with VEGAS for W+jets (left) and t$\bar{\text{t}}$+jets (right).

**Performance**

In Fig. 5, we compare the unweighting efficiencies and relative integration errors for different scenarios and different processes. Throughout all considered processes, we benchmark our trained mappings against the raw mappings combined with and without VEGAS. The shown error bars were obtained by running the integration, including VEGAS optimization if applicable, ten times and taking the mean and standard deviation. All results are shown relative to the VEGAS performance without trained mappings.

We start by considering the W + 3 jets process in the upper left plot of Fig. 5. The trained mappings without additional VEGAS optimization outperform the raw phase-space mapping but are a bit worse than the VEGAS optimized mappings. This is because the number of trainable parameters of our bilinear flow is quite small as it is shared among multiple channels and building blocks, see Tab. 1. In contrast, VEGAS builds an independent grid for each channel and phase-space direction, resulting in more than 30k optimized parameters. When combining our trained mapping with VEGAS, we achieve the best performance in the W + 3 jets scenario, with an improvement factor of up to 1.5. For the t$\bar{\text{t}}$ + 2 jets scenario in the upper right plot, the story is the same.

Next, we consider the same processes but with an additional jet in the final state. The results for different scenarios are shown in the lower two plots in Fig. 5. Again, we consider the mapping that has been trained on the W+3 jets process and evaluate it on the W+4 jets process without further training. We find that the pre-trained mappings are very close in performance to the VEGAS benchmark, without any specific optimization on the W + 4 jets process. Like before, when additionally combining with VEGAS we outperform our untrained phase-space

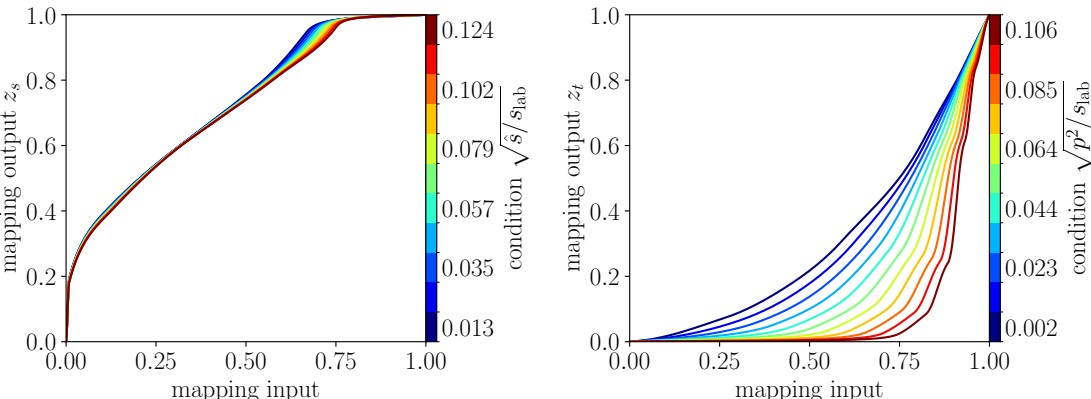

Figure 6: Mappings learned by the bilinear spline flow for W+3 jets. Left panel: Learned mapping for the time-like invariant for massless propagators, conditional on the partonic CM energy $\sqrt{\hat{s}}$. Right panel: Learned mapping for the $t$-invariant in $2 \to 2$ scatterings, conditional on the scattering CM energy $\sqrt{p^2}$.

mappings. If we directly train our mappings on $W + 4$ jets, we immediately outperform our untrained benchmark mappings even without further optimizing with VEGAS. When combining the trained mappings with an additional VEGAS optimization, we achieve an improvement factor of up to 2 for the $W + 4$ jets process.

Again, when turning to the $t\bar{t} + 3$ jets scenario, we observe the same behavior. This indicates that our trainable mappings work well and are capable of generalizing from one process to another process with an additional final state jet. This means our trainable bilinear flow represents the smallest foundation model possible. We note that going even one step further by pre-training our bilinear flows on $W + 2$ jets and $t\bar{t} + 1$ jets, respectively, does not generalize well to higher multiplicities as these low-multiplicity processes are too simple to encode all the necessary information.

**Explainability**

Another benefit of using our bilinear-flow-enhanced mappings is the possibility to understand and interpret the learned correlations. As an example, we consider the learned transformation for the $W+3$ jets process in Fig. 6. Both plots show a learned transformation of an input of one of the phase-space blocks conditioned on some physical features relevant to that component. In the left panel of Fig. 6, we consider the learned transformation for a massless propagator conditional on the partonic CM energy $\sqrt{\hat{s}}$. We can see that the overall shape of the mapping deviates from the flat mapping, being slightly bulged upwards. This means that our fixed choice of $\nu = 1.4$ was slightly too large, indicating stronger pole cancellations in the collinear limit. Further, the mapping tends to avoid $z_s < 0.2$ and hence avoiding to sample the $s_{\min}$ region due to $p_T$ cuts on the final state jets. In contrast, the mapping favors sampling into $s \approx s_{\max}$ stemming from momentum conservation in dominating integration channels containing only $s$-channels. This means, we possibly need to allow for a more flexible optimization of the time-like invariants depending on the underlying topology or the linked particle id. On top of this overall correlation, we can also look into the dependence on the $\sqrt{\hat{s}}$. The condition is varied between 2.5% and 97.5% of the quantile from the distribution of values that this block sees during event generation. We can observe that varying $\sqrt{\hat{s}}$ only has a very small effect on the mapping of the $s$-invariant, indicating a small correlation.

For the right panel of Fig. 6, we consider the learned mapping for the $t$-invariant in a $2 \to 2$ scattering block conditioned on the CM energy $\sqrt{p^2}$ of that $2 \to 2$ scattering. In this case, varying $\sqrt{p^2}$ has a large influence on the optimal $t$-invariant mapping. This means that for larger $p^2$, the mapping tends to sample smaller $z_t$, which is physically linked to smaller scattering angles $\theta^*$ in the center-of-mass system of the $2 \to 2$ scattering block and thus prefers very forward scattering.

## 5 Outlook

Modern machine learning is a promising path to improve the critical multi-purpose event generators to the speed and precision level required by the HL-LHC. For MADGRAPH, the MAD-NIS [11, 12] project has shown that significant gains can be realized by implementing multi-channel importance sampling using modern neural networks.

An exciting and equally promising new approach is differentiable programming applied to event generation [104]. We have tested the potential gains from a differentiable MADNIS for two setups. First, we have developed a fully differentiable combination of matrix element, phase space, and parton densities to use derivative information for optimal integration and sampling. While this setup works well, we have not found significant improvements over the established MADNIS methodology.

As a second and more lightweight use of differentiable code, we have developed a new, modular, and differentiable phase-space mapping. This MADNIS-Lite approach factorizes the phase space into differentiable standard blocks, each consisting of a physics-inspired mapping and a learnable bilinear spline flow. These blocks have the great advantage of being economical and generalizable, so they are much easier to train and use than the full MADNIS framework. For a set of benchmark processes, they show great promise and low computational cost relative to the full MADNIS.

Altogether, this allows us to combine three strategies for future MADGRAPH releases: (i) the standard multi-channel importance sampling using VEGAS techniques provides fast results whenever the integrand factorizes at least approximately; (ii) for more complex Feynman diagrams with factorizing physics structures, which might not be easily mapped on individual phase space directions, MADNIS-Lite provides efficient and fast ML-integration and sampling; (iii) for the highest precision and general matrix elements, including, for instance, gauge cancellations between Feynman diagrams, MADNIS leads to significant improvements over established methods. In combination, these methods provide the ML backbone for optimal event generation at the HL-LHC.

## Acknowledgments

First, we would like to thank Michael Kagan and Lukas Heinrich for continuous encouragement by asking all the right questions many times.

**Funding information**   OM and RW acknowledge support by FRS-FNRS (Belgian National Scientific Research Fund) IISN projects 4.4503.16. TP and TH are supported by the Deutsche Forschungsgemeinschaft (DFG, German Research Foundation) under grant 396021762 – TRR 257 *Particle Physics Phenomenology after the Higgs Discovery*. TH is funded by the Carl-Zeiss-Stiftung through the project *Model-Based AI: Physical Models and Deep Learning for Imaging and Cancer Treatment*. This research is supported by the Deutsche Forschungsgemeinschaft (DFG, German Research Foundation) through Germany's Excellence Strategy EXC 2181/1 – 390900948 (the *Heidelberg STRUCTURES Excellence Cluster*).

The authors acknowledge support by the state of Baden-Württemberg through bwHPC and the German Research Foundation (DFG) through grant no INST 39/963-1 FUGG (bwForCluster NEMO).

# A  Hyperparameters

Table 2: MADNIS hyperparameters used in Sec. 3.

| Parameter | Value |
|---|---|
| Optimizer | Adam [128] |
| Learning rate | 0.001 |
| LR schedule | exponential |
| Final learning rate | 0.0001 |
| Batch size | 1024 |
| Training length | 10k batches |
| Permutations | Logarithmic decomposition [16] |
| Number of coupling blocks | $2\lceil\log_2 D\rceil = 6$ |
| Coupling transformation | RQ splines [129] |
| Subnet hidden nodes | 32 |
| Subnet depth | 3 |
| Activation function | leaky ReLU |

Table 3: MADNIS-Lite and VEGAS hyperparameters used in Sec. 4.

| Parameter | Value |
|---|---|
| Optimizer | Adam |
| Learning rate | 0.01 |
| LR schedule | exponential |
| Final learning rate | 0.001 |
| Batch size | $\min(200 \cdot n_c^{0.8}, 10000)$ |
| Buffered training gain [11] | 6 |
| Training length | 7.8k batches |
| Uniform training fraction [12] | 0.1 |
| RQ spline bins | 6 |
| VEGAS iterations | 7 |
| VEGAS bins | 64 |
| VEGAS samples per iteration | 20k |
| VEGAS damping $\alpha$ | 0.7 |

# B  Analytic loss function gradients

To exemplify the properties of the forward and inverse loss, we consider a simple 1-dimensional target function $f$ as

$$f(x) = \frac{1}{\sqrt{2\pi}} \exp\left(-\frac{x^2}{2}\right). \tag{B.1}$$

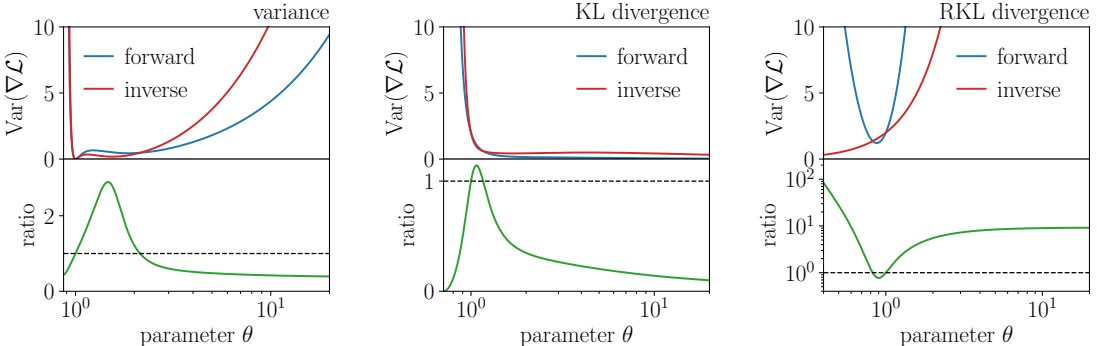

Figure 7: Analytic solutions for the variance of the gradients of different loss functions.

The latent distribution is given as a standard normal distribution as

$$p_0(z) = \frac{1}{\sqrt{2\pi}} \exp\left(-\frac{z^2}{2}\right). \tag{B.2}$$

We now assume a simple trainable mapping (1D flow if you want), given by

$$x \equiv \overline{G}_\theta(z) = \theta \cdot z \quad \longleftrightarrow \quad z \equiv G_\theta(x) = \frac{z}{\theta}, \tag{B.3}$$

inducing the Jacobian determinants

$$\left|\frac{\partial G_\theta(x)}{\partial x}\right| = \frac{1}{\theta} \quad \longleftrightarrow \quad \left|\frac{\partial \overline{G}_\theta(z)}{\partial z}\right| = \theta. \tag{B.4}$$

Combining this with the prior sample density yields the overall importance sampling (pseudo) density

$$g_\theta(x) = p_0(G_\theta(x)) \left|\frac{\partial G_\theta(x)}{\partial x}\right| = \frac{1}{\sqrt{2\pi\theta^2}} \exp\left(-\frac{x^2}{2\theta^2}\right), \tag{B.5}$$

$$\overline{g}_\theta(z) = p_0(z) \left|\frac{\partial \overline{G}_\theta(z)}{\partial z}\right|^{-1} = \frac{1}{\sqrt{2\pi\theta^2}} \exp\left(-\frac{z^2}{2}\right), \quad \text{with} \quad \overline{g}_\theta(G_\theta(x)) = g_\theta(x). \tag{B.6}$$

In this case, the integral of $f$ can be calculated as

$$I = \int \mathrm{d}x \, f(x) = \int \mathrm{d}x \, g_\theta(x) \frac{f(x)}{g_\theta(x)} = \left\langle \frac{g_\theta(x)}{q(x)} \frac{f(x)}{g_\theta(x)} \right\rangle_{x \sim q(x)} = 1. \tag{B.7}$$

**Variance loss**

While the integral does not change, the variance of the integrand is given by

$$\begin{aligned}
\mathcal{L}_{\mathrm{inv}}^{\mathrm{fw}} = \mathcal{L}_{\mathrm{var}}^{\mathrm{inv}} &= \left\langle \frac{g_\theta(x)}{q(x)} \left(\frac{f(x)}{g_\theta(x)} - 1\right)^2 \right\rangle_{x \sim q(x)} \\
&= \int \mathrm{d}x \, g_\theta(x) \left(\frac{f(x)}{g_\theta(x)} - 1\right)^2 \\
&= \frac{\theta^2}{\sqrt{2\theta^2 - 1}} - 1, \quad \text{for} \quad \theta > \frac{1}{\sqrt{2}},
\end{aligned} \tag{B.8}$$

which is exactly zero for the expected value of $\theta = 1$. Next, we can calculate the expectation value of the gradient of the integrand with respect to $\theta$, as this quantity is used during optimization. This yields

$$\nabla_\theta \mathcal{L}_{\text{var}}^{\text{fw}} = \nabla_\theta \mathcal{L}_{\text{var}}^{\text{inv}} = \frac{2\theta(\theta^2 - 1)}{(2\theta^2 - 1)^{3/2}}. \tag{B.9}$$

On the other hand, the variance of the gradient is given by

$$
\begin{aligned}
\text{Var}_{x \sim q(x)} &= \left\langle \left( \nabla_\theta \frac{g_\theta(x)}{q(x)} \left( \frac{f(x)}{g_\theta(x)} - 1 \right)^2 - \nabla_\theta \mathcal{L}_{\text{var}}^{\text{fw}} \right)^2 \right\rangle_{x \sim q(x)} \\
&= \int \mathrm{d}x\, q(x) \left( \nabla_\theta \frac{g_\theta(x)}{q(x)} \left( \frac{f(x)}{g_\theta(x)} - 1 \right)^2 - \nabla_\theta \mathcal{L}_{\text{var}}^{\text{fw}} \right)^2 \\
&= -\frac{1}{4} \left( 2 - \frac{8}{\theta^2} - \frac{2}{(2\theta^2 - 1)^2} + \frac{2}{(2\theta^2 - 1)^3} + \frac{24}{(2\theta^2 - 1)^{5/2}} \right. \\
&\quad - \frac{16}{(2\theta^2 - 1)^{3/2}} - \frac{2}{2\theta^2 - 1} + \frac{8}{(2\theta^2 - 1)^{1/2}} \\
&\quad \left. - \frac{9}{(4\theta^2 - 3)^{5/2}} + \frac{3}{(4\theta^2 - 3)^{3/2}} - \frac{1}{(4\theta^2 - 3)^{1/2}} - \sqrt{4\theta^2 - 3} \right).
\end{aligned}
\tag{B.10}
$$

However, for the inverse training, we obtain the variance of the gradient as

$$
\begin{aligned}
\text{Var}_{z \sim p_0(z)} &= \left\langle \left( \nabla_\theta \left( \frac{f(\overline{G}_\theta(z))}{\overline{g}_\theta(z)} - 1 \right)^2 - \nabla_\theta \mathcal{L}_{\text{var}}^{\text{inv}} \right)^2 \right\rangle_{z \sim p_0(z)} \\
&= \int \mathrm{d}z\, p_0(z) \left( \nabla_\theta \left( \frac{f(\overline{G}_\theta(z))}{\overline{g}_\theta(z)} - 1 \right)^2 - \nabla_\theta \mathcal{L}_{\text{var}}^{\text{inv}} \right)^2 \\
&= -\frac{1}{16} \left( 8 - \frac{8}{(2\theta^2 - 1)^2} + \frac{8}{(2\theta^2 - 1)^3} - \frac{48}{(2\theta^2 - 1)^{5/2}} \right. \\
&\quad - \frac{32}{(2\theta^2 - 1)^{3/2}} - \frac{8}{2\theta^2 - 1} - \frac{48}{(2\theta^2 - 1)^{1/2}} \\
&\quad - \frac{81}{(4\theta^2 - 3)^{5/2}} - \frac{9}{(4\theta^2 - 3)^{3/2}} - \frac{27}{(4\theta^2 - 3)^{1/2}} \\
&\quad \left. - 11\sqrt{4\theta^2 - 3} + \frac{256\theta(3\theta^4 - 4\theta^2 + 2)}{(3\theta^2 - 2)^{5/2}} \right).
\end{aligned}
\tag{B.11}
$$

**KL loss**

For the expectation value of the gradient, we obtain for both directions

$$\nabla_\theta \mathcal{L}_{\text{KL}}^{\text{fw}} = \nabla_\theta \mathcal{L}_{\text{KL}}^{\text{inv}} = \frac{\theta^2 - 1}{\theta^3}. \tag{B.12}$$

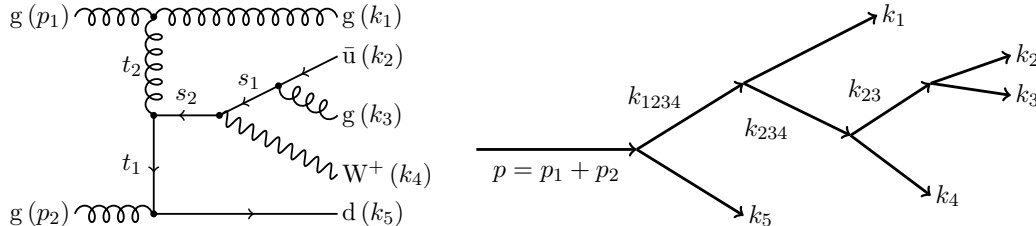

Figure 8: An example Feynman diagram contributing to the $gg \to W^+\bar{u}dgg$ process (left) and an illustration of the corresponding phase-space parametrization (right).

While for the variance, we obtain

$$\text{Var}\left(\nabla_\theta \mathcal{L}_{\text{KL}}^{\text{fw}}\right) = -\frac{1}{\theta^6} + \frac{2}{\theta^4} - \frac{1}{\theta^2} + \frac{4\theta^4 - 8\theta^2 + 6}{(2\theta^2 - 1)^{5/2}} \tag{B.13}$$

$$\begin{aligned}
\text{Var}\left(\nabla_\theta \mathcal{L}_{\text{KL}}^{\text{inv}}\right) = \frac{-1}{4\theta^6(2\theta^2 - 1)^{9/2}} \Big( & \sqrt{2\theta^2 - 1}\left(4 - 40\theta^2 + 164\theta^4\right) \\
& - \theta^6\Big[3 + 49\theta^8 + 352\sqrt{2\theta^2 - 1} - 4\theta^6\left(27 + 16\sqrt{2\theta^2 - 1}\right) \\
& + 8\theta^4\left(9 + 32\sqrt{2\theta^2 - 1}\right) - 8\theta^2\left(1 + 52\sqrt{2\theta^2 - 1}\right)\Big] \\
& - 4\theta^6 \log(\theta)\Big[1 - 11\theta^2 + 27\theta^4 - 23\theta^6 + 10\theta^8 \\
& + (1 - 2\theta^2)^2(1 - 2\theta^2 + 3\theta^4)\log(\theta)\Big]\Big) \, .
\end{aligned} \tag{B.14}$$

**RKL loss**

For the expectation value of the gradient, we obtain for both directions

$$\nabla_\theta \mathcal{L}_{\text{RKL}}^{\text{fw}} = \nabla_\theta \mathcal{L}_{\text{RKL}}^{\text{inv}} = \frac{\theta^2 - 1}{\theta} \, . \tag{B.15}$$

While for the variance of the gradient, we obtain

$$\text{Var}\left(\nabla_\theta \mathcal{L}_{\text{RKL}}^{\text{fw}}\right) = \frac{21 - 54\theta^2 + 37\theta^4 + 4\log(\theta)(3 - 5\theta^2 + \log\theta)}{2\theta^2} \, , \tag{B.16}$$

$$\text{Var}\left(\nabla_\theta \mathcal{L}_{\text{RKL}}^{\text{inv}}\right) = 2\theta^2 \, . \tag{B.17}$$

In Fig. 7, we illustrate the gradient variances for the forward and inverse training for the different loss functions. The upper panels show the absolute gradient variance, while the lower panel shows the ratio between the forward and inverse directions. A ratio of $r > 1$ means the gradient variance of the forward training is larger than the gradient variance of the inverse training. For the variance loss and KL divergence, we can observe that the forward training yields the more stable training. Only in parameter regions around the optimal value, i.e. $\theta \approx \theta_{\text{opt}} = 1$ the inverse training is more stable. In contrast, for the RKL divergence, the picture changes and the inverse loss gives less noisy gradients.

## C  Explicit channel mapping

As an example, we consider $W + 4$ jets production

$$gg \to W^+\bar{u}dgg \, . \tag{C.1}$$

In particular, we investigate the Feynman diagram of Fig. 8, because it involves all types of phase-space blocks introduced in Sec. 4. We define

$$k_{23} = k_2 + k_3, \qquad k_{234} = k_2 + k_3 + k_4, \qquad k_{1234} = k_1 + k_2 + k_3 + k_4,$$
$$q_1 = p_1 - k_1, \qquad q_2 = p_2 - k_5, \qquad p = p_1 + p_2. \tag{C.2}$$

The infrared and collinear singularities are excluded by a lower cut on $k_{23}^2 > k_{23,\text{min}}^2$. The phase-space integral

$$\int d\Phi^{2\to5}\Big|_{\text{Fig.8}} = \int_{k_{23,\text{min}}^2}^{\hat{s}} dk_{23}^2 \int_{k_{23}^2}^{\hat{s}} dk_{234}^2 \int_{k_{234}^2}^{\hat{s}} dk_{1234}^2 \int d\Phi^{2\to2}(p_1, p_2; k_{1234}^2, k_5^2) \tag{C.3}$$
$$\times \int d\Phi^{2\to2}(p_1, q_2; k_1^2, k_{234}^2) \int d\Phi^{1\to2}(k_{234}; k_{23}^2, k_4^2) \int d\Phi^{1\to2}(k_{23}; k_2^2, k_3^2),$$

is decomposed into two $2 \to 2$ scattering processes and two decays. The intermediate particles of this decomposition are the virtual particles with momenta $k_{23}$ and $k_{234}$, and an additional pseudo particle with momentum $k_{1234}$. First, we perform the luminosity sampling according to Eq.(49) and obtain

$$\hat{s} = s_{\text{lab}}^{z_0} \hat{s}_{\text{min}}^{1-z_0}, \quad \text{with} \quad \hat{s}_{\text{min}} = (M_{\text{W}} + k_{23,\text{min}})^2,$$
$$\xi = \frac{1}{2} \log \frac{x_1}{x_2}, \tag{C.4}$$

needed to define $p_{1,2}$ and perform the boost into the proton-proton rest frame. Next, the invariant masses of the external particles of the scattering processes and particle decays have to be determined

$$k_{23}^2 = \overline{G}_{\text{prop}}^{\nu_1}(z_1, 0, k_{23,\text{min}}^2, \hat{s}),$$
$$k_{234}^2 = \overline{G}_{\text{prop}}^{\nu_2}(z_2, 0, k_{23}^2, \hat{s}), \tag{C.5}$$
$$k_{1234}^2 = \overline{G}_{\text{prop}}^{\nu=0}(z_3, 0, k_{234}^2, \hat{s}) \equiv z_3(\hat{s} - k_{234}^2) + k_{234}^2.$$

While the time-like invariants $s_1 = k_{23}^2$ and $s_2 = k_{234}^2$ correspond to massless propagators in the diagram, $s_3 = k_{1234}^2$ is the squared CM energy of the first $2 \to 2$ scattering ($t_1$) and belongs to a pseudo particle. Consequently, $s_3$ is sampled flat. As a next step, the final-state momenta are calculated step-by-step:

(i) $p_1 + p_2 \to k_{1234} + k_5$:
The initial-state gluons transform into the final-state d-quark and a pseudo particle with momentum $k_{1234}$. The invariant mass of the d-quark propagator is given by

$$|q_1^2| = \overline{G}_{\text{prop}}^{\nu_3}(z_4, 0, 0, \hat{s} - k_{1234}^2), \tag{C.6}$$

where the boundaries are taken from Eq.(43).

(ii) $p_1 + q_2 \to k_1 + k_{234}$:
The incoming particles are the initial-state gluon and the incoming virtual d-quark. We further note that $q_2^2 = (p_1 - k_5)^2 = t_1$. The invariant mass of the gluon propagator then reads

$$|q_2^2| = \overline{G}_{\text{prop}}^{\nu_4}(z_5, 0, 0, (k_{1234}^2 - k_{234}^2)(k_{1234}^2 - t_1)/k_{1234}^2), \tag{C.7}$$

where the boundaries are calculated again from Eq.(43). This fixes the momenta $k_1$ and $k_{234}$ of the outgoing gluon and the virtual d-quark, respectively.

(iii) $k_{234} \to k_{23} + k_4$:
The virtual $\bar{\text{d}}$-quark decays isotropically into the final-state W-boson ($k_4$) and the virtual $\bar{\text{u}}$-quark ($k_{23}$).

(iv) $k_{23} \to k_2 + k_3$:
Finally, the virtual $\bar{\text{u}}$-quark decays isotropically into the final-state $\bar{\text{u}}$-quark ($k_2$) and a gluon ($k_3$).

The total phase-space density is then given by

$$
\begin{aligned}
g_{\text{tot}} = {} & g_{\text{lumi}}(\tau, \tau_{\min}) g_{\text{prop}}^{\nu_1}(k_{23}^2, 0, k_{23,\min}^2, \hat{s}) \, g_{\text{prop}}^{\nu_2}(k_{234}^2, 0, k_{23}^2, \hat{s}) \, g_{\text{prop}}^{\nu=0}(k_{1234}^2, 0, k_{234}^2, \hat{s}) \\
& \times g_{2 \to 2}(\hat{s}, 0, 0, t_1, 0, \nu_3, 0, \hat{s} - k_{1234}^2) \\
& \times g_{2 \to 2}(k_{1234}^2, 0, t_1, t_2, 0, \nu_4, 0, (k_{1234}^2 - k_{234}^2)(k_{1234}^2 - t_1)/k_{1234}^2) \\
& \times g_{\text{decay}}(k_{234}^2, k_{23}^2, 0) \, g_{\text{decay}}(k_{23}^2, 0, 0),
\end{aligned}
\tag{C.8}
$$

which includes all propagators of this diagram.

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
