# Peer review of "Differentiable MadNIS-Lite"

_SciPost Physics, doi:SciPost Phys. 18, 017 (2025)_

## Round 1 · Referee Report · Anonymous (Referee 1) · 2024-10-7

Strengths

  • The manuscript delivers a more in-depth study on the potential of using differential programming for matrix elements, PDF evaluation, and in particular phase-space generation, than has previously been available.

  • Studies like these can discover and explore new avenues in the development of efficient event generators for the HL-LHC and other future colliders, which come with a much increased requirement on simulated event sample sizes; here, first steps are shown towards using differential programming as part of the toolbox of future efficient event generation frameworks. Moreover, differential programming is helpful for understanding parameter dependencies and estimate uncertainties and can become an essential tool for simulations frameworks such as event generators.

Weaknesses

  • The introduction makes too strong statements with respect to the requirements and future development of the techniques as used in the field (see "Requested changes"); in particular, "modern machine learning methods" are implied to be the only way to deliver the required sample sizes, ignoring non-ML developments that already deliver significant speed-ups (such as "old-school" performance improvements as in 2209.00843, or the various projects porting event generation to GPU). If the authors disagree, they would need to back up their statements with references supporting them.

  • The contents of Sec. 2 is not always in itself clear without referring to previous publications of the collaboration (see "Requested changes").

  • In terms of the figures of merit laid out (standard deviation and unweighting efficiency), a breakthrough over existing techniques is not achieved, but modest factors of 1.5--2 are shown with respect to previous implementations in MadGraph. A comparison with tools beyond MadGraph is missing altogether; a lower computational cost of MadNIS-lite compared to the full MadNIS technique is claimed in the conclusions, but not shown.

  • The quality of the baseline VEGAS-optimized results is not specified; i.e. what steps have been made to ensure that the VEGAS optimization is fully converged, in particular with respect to the unweighting efficiency, or to ensure that the number of bins is appropriate etc. These questions are particularly important because Table 3 seems to indicate that at least ten times more points are used to train the new approaches, compared to the number of points used to train the VEGAS-only benchmark.

Report

The authors deliver a timely study of potential advantages of using differentiable programming for event generation and in particular phase-space generation. It goes beyond previous proof-of-principles by studying potential advantages of the technique for optimizing phase-space generation. The study methods and results are explained in a detailed and reproducible way.

Some of the identified weaknesses should be addressed by a minor revision, see "Requested changes".

Otherwise, the journal's acceptance criteria are met, although the author's claim that the manuscript "[details] a groundbreaking theoretical/experimental/computational discovery" is a stretch, given that known methods are applied to a previously identified problem, a proof-of-principle study of applying the method already exists, and given that the results only suggest incremental improvements at this point. The study has great merits, and there is no need to embellish it.

Requested changes

  1. Introduction

  2. 1.1 In the first paragraph, the authors claim that "we will have to rely on modern machine learning (ML) [4, 5] to significantly improve the speed and the precision of these simulations". However, this statement is not supported in [4, 5], nor is it self-evident. Other methods that deliver or potentially can deliver large speed-ups, e.g. profiling and addressing bottlenecks in existing codebases as in arXiv:2209.00843, or modifications/adaptations of the relevant algorithms for optimized use on large HPC clusters or to profit from hardware accelerators such as GPU are not mentioned here, and might themselves be sufficient e.g. for the requirements of the HL-LHC. While I leave it to the authors whether they want to present their work in this wider context, I would ask to at least tone down the statement and just write something like "modern machine learning methods can help us to […]" instead.

  3. 1.2 Similarly, still in the first paragraph, the claim that "modern neural networks will transform" many of the components of the event generation pipeline is much too strong. I would ask the authors to tone it down. I understand that the purpose of the sentence is mostly to list relevant related work, but the current formulation wrongly implies that a groundbreaking paradigm shift induced by using neural networks is going on in the field. While the authors might argue otherwise, a claim like that should clearly not be done en passant, but should be properly argued.

  4. 1.3 In the second paragraph, the authors write that generative networks are "currently trained on and amplifying simulated training data". This formulation implies that these techniques are actively used for the production pipeline e.g. by the LHC experiments. To my knowledge, they are not. I would ask the authors to remove "currently" to avoid this possible misunderstanding.

  5. 1.4 In the third paragraph, the authors write "A proof of principle has been delivered for differentiable matrix elements [99], but the same methods are used for differentiable detector design […]" Should the use of the conjunction "but" here imply that only [99] is a proof of principle, and the other studies cited [100-103] are not, i.e. ready for production use? I believe not, given that at least some of these again have signifiers like "towards" or "first steps" in their titles and/or abstracts. Perhaps the authors could refine their formulation here.

  6. MadNIS basics

  7. 2.1 Regarding the two equations in Eq. (8), the authors seem to indicate that these are two possible prior assumptions. Are they? But then the authors should not use the same symbol, since they are not equal (perhaps the authors accidentally used the same superscript?) Also, while I understand that this section is just meant as a quick review, please make clear to introduce all symbols that appear in Eq. (8). Currently, this is not the case.

  8. 2.2 In Eq. (10), theta seems to appear for the first time, but it is not introduced as a new set of network parameters if I'm not mistaken. I would ask the authors to introduce theta in case I have not overlooked it.

  9. Differentiable phase space — MADNIS-Lite

  10. 4.1 There is a typo in the beginning of Sec. 4.2. "For typical a […]" Swap "typical" and "a".

  11. I would ask the authors to provide details on the VEGAS baseline. How did the authors ensure that the optimization is fully converged (or trained with a similar computational effort)? As the unweighting efficiency depends on the tail of the weight distribution, further optimisation can still make a large difference even if e.g. the integral itself has already converged nicely. Table 3 seems to indicate that the authors use at least ten more points for training the MadNIS-Lite parameters compared to what the authors use to optimize the baseline VEGAS grids (but perhaps I am misunderstanding the numbers given in the table). This becomes even more puzzling considering that the number of VEGAS parameters is much higher than the number of MadNIS-Lite parameters, as the authors state in the "Implementation" paragraph. More details would help to understand better how the baseline the authors compare against is constituted, which is important to fully appreciate their results.

  12. Outlook

  13. 5.1 In the conclusions on MadNIS-Lite, the authors write that "for a set of benchmark processes, they show great promise and low computational cost." This is a bit vague, in particular given that there is no direct comparison with the full MadNIS results given in the draft, and because there is no quantifiable data on computational cost. I would ask the authors to write more explicitly what the conclusions for MadNIS-Lite are, given the results in the draft. This would also further contextualize the outlook on the MadGraph developments in the last paragraph.

Recommendation

Ask for minor revision

  • validity: high
  • significance: good
  • originality: good
  • clarity: high
  • formatting: perfect
  • grammar: perfect

Author:  Ramon Winterhalder  on 2024-11-19  [id 4964]

(in reply to Report 1 on 2024-10-07)
Category:
answer to question

1.1 We now also mention other endeavors to address bottlenecks based on improved and/or hardware-accelerated code.

1.2 We do not believe this is too strong of a claim, given all the progress in recent years.

1.3 We agree and changed the wording

1.4 In this case, "but" refers to the fact that only the first paper [99] has worked on differentiable matrix elements, while the others worked on other aspects of the simulation chain.}

2.1 We now distinguish both priors by having different superscripts. As we otherwise refer to both the original papers where these bases have been introduced and the last MadNIS paper, we refrain from giving even more details here.

2.2 Done and fixed in the new version.

4.1 Fixed in the new version

4.2 We trained VEGAS with the same setup as in the MadNIS Reloaded paper before, where we found that training VEGAS with much much longer iterations and/or samples does not further improve the unweighting efficiency. We checked that we had reached a plateau in performance and subsequently used one of these training setups. Further, while indeed MadNIS-lite is trained on more samples than VEGAS even though the number of parameters is smaller, this does not mean VEGAS is not converged. The important difference is the way of training. Even though MadNIS-lite has fewer parameters, it is trained via stochastic gradient descent, which needs more training updates and data to converge properly. In contrast, the training algorithm of VEGAS is much simpler and thus converges much faster. Note that this is why we usually pre-train with VEGAS and only refine the normalizing flow on the remaining correlations.

5.1 We did not include the MadNIS results for comparison, as (i) we show the same examples as in the last paper where all MadNIS results are available, and (ii) the main purpose of this paper was not to beat the full MadNIS but rather show how to develop a more economical and interpretable framework. We did not include numbers for absolute computational cost, but the sentence in the conclusion refers to the full MadNIS approach, which relies on a much larger network and consequently requires more computing power. We have now refined this sentence in the conclusion to make it clearer.

---

## Round 1 · Referee Report · Anonymous (Referee 2) · 2024-10-8

Strengths

Very innovative study of using ML-based methods to sample phase space for complicated processes in particle physics. The new ingredient is the decomposability into kinematic subfeatures of phase space and the differentiability of these building blocks, which opens the possibility to learn the dependence of phase space features, matrix elements and PDFs on certain parameters of the problem.

Weaknesses

As of yet, this is still a proof-of-concept and a thorough study over the full zoo of QCD and electroweak processes in LHC (and maybe other collider) setups needs to follow. It is not clear that the number of building blocks is sufficient to describe each and every kinematic features in all setups. Potentially the dependence on "commercial" tools like pyTorch or TensorFlow might be considered a risk.

Report

The authors present a very interesting proof-of-concept implementation of ML-based phase-space sampling, combined with traditional adaptive MC integration methods and - as a new ingredient - differentiable building blocks for the components of the learning. I find the results presented very interesting, and even if clearly that is just a first step to potentially replace well-established PS sampling algorithms in the future, the conceptual work in this paper deserves publication in a journal like SciPost. Clearly, there is a lot of potential in that framework, which is easily generalizable to more complicated processes and setups, which to me is the main strength of the paper. The two examples used in the manuscript are DY + 3/4 jets or ttbar + 2/3 jets, so processes that are dominated by a single resonance or pair production with additional jet radiation. It would be highly interesting to see this setup being applied to multi-heavy-resonance production (like VVV, V=W,Z) or VBF/VBS-like processes.

One point that to me did not become fully clear is how much of the multi-channel setup of the processes studied in the paper within the MadNIS-Lite framework is covered by the multi-channel version of the VEGAS algorith, and what is covered by the trained spline parameterization. Particularly (maybe I overlooked that aspect), are channel weights among the training parameters in MadNIS-Lite or not, and are their correlations between different channels that influence the gain in performance during the training?

The authors chose to implement their building blocks into pre-defined framework like pyTorch or Tensorflow, and I am curious to get to know whether this poses any form of constraints on the implementation or also on the maintenance of such a framework for one-and-a-half decades of HL-LHC to come.

One question on the end of the first part of Sec. 3.2, below Fig. 2 on 8: to me it is not obvious why it is not possible to define a multi-channel generalizatiojn of the KL-divergence. The authors mention that this seems intrinsically possible only for the variance loss, but somehow I stupidly fail to see why.

In the discussion of Fig. 6 in the last part of Sec. 4 the authors discuss a point I was wondering about when looking up the W+3jet PS parameterization in App. C: clearly for infrared QCD radiation the outmost minimum of shat is M_W^2, however, the authors do apply a cut to cure the IR singularity on k_23^2, so shouldn't the minimum of shat not rather be given by M_W^2 + k_23,min^2? The authors are mentioning this fact by the avoidance of the region < 0.2 in the sampling of the time-like invariant.

On Appendix B, I would have the question how general the authors consider their findings on the stability of the trainings comparing variance, Kullbaeck-Leibler and reverse KL loss, especially when going from analytic toy ensembles to LHC physics processes?

Requested changes

Some small things that I found and believe to be typos: - In App. C there seems to be a conflict in notation between q_1 in Eqs. (69) vs. item (ii) above Eq. (74). q_1 should be either p1 - k5 or p1 - k1. In Eq. (70) the third integration variable should be k_{1234}^2

Recommendation

Publish (easily meets expectations and criteria for this Journal; among top 50%)

  • validity: high
  • significance: high
  • originality: top
  • clarity: high
  • formatting: excellent
  • grammar: excellent

Author:  Ramon Winterhalder  on 2024-11-19  [id 4965]

(in reply to Report 2 on 2024-10-08)
Category:
answer to question

One point that to me did not become fully clear is how much of the multi-channel setup of the processes studied in the paper within the MadNIS-Lite framework is covered by the multi-channel version of the VEGAS algorithm, and what is covered by the trained spline parameterization. Particularly (maybe I overlooked that aspect), are channel weights among the training parameters in MadNIS-Lite or not, and are their correlations between different channels that influence the gain in performance during the training?

-> For what we call VEGAS and MadNIS-lite, the channel weights are the standard weights as defined in MadGraph. As mentioned on p.14, they are not trainable parameters. Only in the full MadNIS setup the channel weights become trainable.

The authors chose to implement their building blocks into pre-defined framework like pyTorch or Tensorflow, and I am curious to get to know whether this poses any form of constraints on the implementation or also on the maintenance of such a framework for one-and-a-half decades of HL-LHC to come.

-> For this project and paper, we decided to use PyTorch as it was the simplest way to implement a differentiable integrator that can be easily connected with MadNIS, which is also written in PyTorch. However, there is no conceptual problem with implementing everything instead in C++ or Rust.

One question on the end of the first part of Sec. 3.2, below Fig. 2 on 8: to me it is not obvious why it is not possible to define a multi-channel generalizatiojn of the KL-divergence. The authors mention that this seems intrinsically possible only for the variance loss, but somehow I stupidly fail to see why.

-> In principle, you can also define the KL-divergence in a multi-channel approach, where you simply apply a KL-divergence independently to each channel and then sum over all terms. However, a non-trivial question is how to combine these channels in this sum. Are they all just naively summed with the same weight? Are they weighted according to some fixed weight based on the channel uncertainty? Or is the weight some hyperparameter you want to tune? These questions can only be answered unambiguously when using the Variance loss, where the proper definition of these weights naturally appears in the mathematical definition of the uncertainty. This is not the case when using the KL-divergence.

In the discussion of Fig. 6 in the last part of Sec. 4 the authors discuss a point I was wondering about when looking up the W+3jet PS parameterization in App.~C: clearly for infrared QCD radiation the outmost minimum of shat is $M_W^2$, however, the authors do apply a cut to cure the IR singularity on $k_{23}^2$, so shouldn't the minimum of shat not rather be given by $M_W^2 + k_{23,\min}^2$? The authors are mentioning this fact by the avoidance of the region < 0.2 in the sampling of the time-like invariant.

-> Indeed, we forgot to account for that cut in Eq.(71) and changed the expression for the minimum of shat accordingly.

On Appendix B, I would have the question how general the authors consider their findings on the stability of the trainings comparing variance, Kullbaeck-Leibler and reverse KL loss, especially when going from analytic toy ensembles to LHC physics processes?

-> We first want to emphasize that the results in Appendix B do not serve as mathematical proof but rather help to develop an intuition of why forward/inverse training works better or worse for several loss functions. As we can see in the results in Figure 1 for LHC processes, the general trend that the forward loss is superior persists. However, as we can see in Figure 2, the KL divergence performs best when applied to this specific LHC process. This indicates that depending on the actual problem, different loss functions might perform better. However, again, only the variance loss can be naturally extended to the multi-channel case, as explained above.

Some small things that I found and believe to be typos.

-> Fixed in the new version

---

## Round 2 · Referee Report · Anonymous (Referee 2) · 2024-11-20

Report

The authors have answered all the raised questions of the previous report and have implemented the requested changes or corrections.

Recommendation

Publish (easily meets expectations and criteria for this Journal; among top 50%)

---

## Round 2 · Referee Report · Anonymous (Referee 1) · 2024-11-27

Report

The authors have answered all the raised questions of my previous report and have implemented the requested changes or corrections in a satisfactory way.

Recommendation

Publish (easily meets expectations and criteria for this Journal; among top 50%)

---

## Round 2 · Author Response

We answered all referee questions directly as comments to their report.

---

## Editorial Decision

published